

# Specx: a C++ task-based runtime system for heterogeneous distributed architectures

Paul Cardosi[1,2,3] and Bérenger Bramas[1,2,3]

[1] ICPS Team, ICube, Illkirch, France
[2] CAMUS Team, Inria, Illkirch, France
[3] University of Strasbourg, Illkirch, France

## ABSTRACT

Parallelization is needed everywhere, from laptops and mobile phones to supercomputers. Among parallel programming models, task-based programming has demonstrated a powerful potential and is widely used in high-performance scientific computing. Not only does it allow efficient parallelization across distributed heterogeneous computing nodes, but it also allows for elegant source code structuring by describing hardware-independent algorithms. In this article, we present Specx, a task-based runtime system written in modern C++. Specx supports distributed heterogeneous computing by simultaneously exploiting central processing units (CPUs) and graphics processing units (GPUs) (CUDA/HIP) and incorporating communication into the task graph. We describe the specificities of Specx and demonstrate its potential by running parallel applications.

## INTRODUCTION

Modern computers are increasingly heterogeneous and hierarchically structured, both in terms of memory and parallelization. This is especially visible in the high-performance computing (HPC) environment, where clusters of computing nodes equipped with multi-core central processing units (CPUs) and several graphics processing units (GPUs) are becoming the norm. Programming applications to efficiently use these types of architectures is challenging and requires expertise.

The research community has suggested various runtime systems to help parallelize computational codes. These tools differ on many aspects, including the hardware they target, their ease of use, their performance and their level of abstraction. Some runtime systems have demonstrated flexibility in their use but they are designed for experts, such as StarPU (*Augonnet et al., 2011*). Others provide a modern C++ interface but they do not support as many features as what HPC applications need, such as Taskflow (*Huang et al., 2022*).

In our current study, we describe Specx (/'spɛks/) (https://gitlab.inria.fr/bramas/specx), a runtime system that has been designed with the objective of providing the features of advanced HPC runtime systems, while being easy to use and allowing developers to write modular and easy to maintain applications.

Corresponding author
Bérenger Bramas,
berenger.bramas@inria.fr

The contribution of our work can be summarized as follows:

- We describe the internal organization of Specx, a task-based runtime system written in modern C++.
- We present the key features needed to develop advanced HPC applications, such as scheduler customization, heterogeneous tasks and dynamic worker teams.
- We show that Specx allows developers to write compact C++ code thanks to advanced meta-programming.
- Finally, we demonstrate the performance of Specx on several test cases.

The manuscript is organized as follows. We provide the prerequisites in "Background" and the related work in "Related Work". Then, we describe Specx in "Specx's Features, Design and Implementation", before the performance study in "Performance and Usability Study". Portions of this text were previously published as part of a preprint (https://arxiv.org/abs/2308.15964).

## BACKGROUND

In this section, we briefly describe task-based parallelization and the challenges it faces when computing on heterogeneous architectures.

### Task-based parallelization

Task-based parallelization is a programming model in which the application is broken down into a set of tasks. It relies on the principle that an algorithm can be broken down into interdependent operations, where the output of some tasks is the input of other tasks. The tasks can be executed independently or in parallel and they can be dynamically scheduled to different processing units while ensuring execution coherency. The result can be seen as a direct acyclic graph (DAG) of tasks or simple graph of tasks, where each node is a task and each edge is a dependency. An execution of such a graph will start from the nodes that have no predecessor and continue forward in the graph, ensuring that when a task starts, all its predecessors have completed. The granularity of the tasks, that is, the content in terms of computation, cannot be too fine-grained because the internal management of the graph implies an overhead that must be negligible to ensure good performance (*Tagliavini, Cesarini & Marongiu, 2018*). Therefore, it is usually the developer's responsibility to decide which and how many computations to bundle inside a task. The granularity is then a balance between the degree of parallelism and the runtime system overhead. To reduce the overhead, several researches are conducted to delegate partially or totally the runtime system to the hardware (*Chronaki et al., 2018*).

The dependencies between tasks can be described in various ways. One way is to have the user explicitly connect tasks together. For example, the user might call a function $connect(t_i, t_j)$ to connect tasks $t_i$ and $t_j$. This approach requires the user to manage the coherency and to keep track of the dependencies between tasks, which can be error-prone and complicated between different stages of an application. TaskFlow uses this approach.

Another way is to inform the runtime system about the input/output of tasks and let it take care of the coherency. This approach is more convenient for the users but there are many ways this approach can be implemented. One approach is to use a mechanism like the C++ future to access the result of asynchronous operations. This approach allows the runtime system to track the dependencies between tasks and ensure that they are satisfied without having a view on the input/output. This approach is used by the ORWL runtime system (*Clauss & Gustedt, 2010*).

An alternative is to use sequential task-flow (STF) (*Agullo et al., 2016c*), also called task-based data programming. In this approach, the user describes the tasks and specifies data inputs and outputs for each task. In general, a single thread creates the tasks and submits them to a runtime system while specifying the data dependencies for each task. The runtime system is then able to generate the graph and guarantee that the parallel execution will have the absolute same result as a sequential one. This results in a very compact code with few modifications required to be applied to an existing application, by moving the complexity into the runtime system. Reads after writes and writes after reads occur in the order in which read and write dependencies of tasks are given to the runtime system, that is the sequential task insertion order. This approach has a number of advantages, including:

- A sequential program can be transformed into a parallel equivalent very easily.
- The users do not have to manage the dependencies.
- The accesses can be more precise than read/write and specific properties can be set for the accesses, such as commutativity.
- The tasks can be mapped to a graph, allowing the runtime system to analyze the graph to predict the workload or memory transfers and to make informed decisions.

In our work, we use the STF model.

## Computing on heterogeneous architectures

Heterogeneous computing nodes consist of at least two distinct types of processing units. The most common configuration includes a dual-socket CPU paired with one or several GPUs, each having separate memory nodes (see Fig. 1). However, similar principles apply to other types of processing units as well.

Traditionally, these nodes operate in a pattern where a single CPU thread manages data movement to the device's memory, initiates the computational kernel and waits for its completion before transferring the data back if required. To assist programmers, vendors have introduced unified memory, a mechanism that creates the illusion of a shared memory space between CPUs and GPUs. However, due to potential unpredictability and lack of control, its use remains rare in high-performance computing (HPC). Meanwhile, this usage pattern leaves other CPU cores idle, which is untenable in HPC.

To increase the utilization of processing units, this pattern can be expanded to incorporate multiple CPU threads sharing a single GPU, enabled by mechanisms like streams or queues. This arrangement allows full harnessing of the GPU in terms of

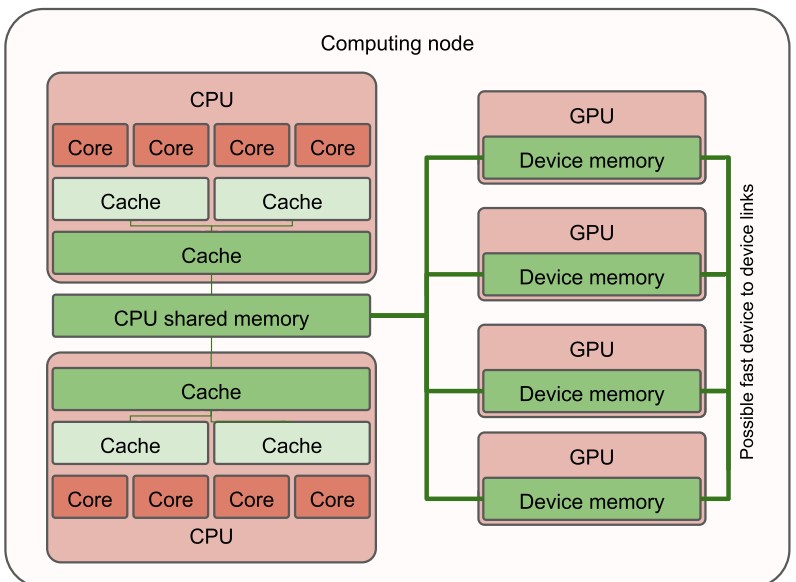

**Figure 1 Simplified view of a heterogeneous computing node with two CPUs and four GPUs.** Multiple nodes can be interconnected *via* a network.

computational capability and memory transfers. However, managing the device's memory becomes increasingly complex, as it becomes crucial to avoid redundant object copying and ensure memory capacity isn't exceeded.

Furthermore, this method introduces the key challenge of balancing heterogeneous computing. Specifically, determining the optimal number of CPU threads per GPU, figuring out how to best use idle CPU cores and where to best execute each application part (CPU or GPU?). In essence, how can we optimally distribute work among all processing units?

Task-based runtime systems aim to resolve these issues. They predominantly manage data transfers between CPUs and GPUs, allocate specific CPU cores to control GPUs, separate these cores from others to allow for concurrent executions and schedule tasks across various types of processing units while considering workload and the most efficient processing unit type.

While these runtime systems provide a means to efficiently execute tasks, the implementation and optimization of the tasks themselves are left to the developer. For example, for the CPU, optimization of a task's code may consist in using vectorization and an efficient memory access pattern. For the GPU, one may use dedicated language extensions, such as CUDA for Nvidia GPUs or HIP for AMD GPUs and optimize GPU kernels using shared memory or warp-level primitives. In addition, there are tools that allow creating portable kernels that can run both on the CPU and the GPU. For example, Kokkos (*Trott et al., 2022*; *Edwards, Trott & Sunderland, 2014*) can be used to implement portable computational kernels for both the CPU and the GPU and to parallelize code using various backends such as HIP, CUDA, OpenMP and HPX. Similarly, SYCL (*Reyes & Lomüller, 2016*) is an abstraction layer that enables writing C++ code for heterogeneous

computing platforms (including, for example, CPUs, GPUs and FPGAs) in the form of a single portable computational kernel.

## RELATED WORK

### Task-based parallelization

The most common task-based programming pattern can be described as a tasks-and-wait scheme, where independent tasks are inserted into a runtime system and a synchronization point allows the program to wait for the tasks to complete. The task model from OpenMP 3 (*OpenMP Architecture Review Board, 2008*; *Ayguadé et al., 2009*) and the task-based programming language Cilk (*Blumofe et al., 1996*) (later extended to Cilk++; *Leiserson, 2009*) and Cilk Plus (https://www.cilkplus.org/) follow this idea. This remains a fork-join model because successive spawn phases of independent tasks (fork) must be explicitly synchronized (join) to ensure correct execution. Therefore, it limits the scalability because of the waiting time and the imbalance between tasks. Of course, developers can increase the degree of parallelism by using multiple sources of tasks that they know are independent. However, such an implementation starts to become a manual management of dependencies, which a modern task-based runtime system is intended to take care of.

This is why there now exist numerous different task-based runtime systems that support dependency management. The most popular ones are implementations of the OpenMP version 4 (*OpenMP Architecture Review Board, 2013*; *Chandra et al., 2001*) standard that defines the additional pragma keyword *depend* to inform the runtime system about the type of data accesses performed by the tasks. However, using pragmas, in general, is tedious when a task has hundreds of dependencies or when the number of dependencies is only known at runtime. This can lead to ugly and error-prone code. In addition, as OpenMP is a standard, it is upgraded slowly to ensure backward compatibility. Moreover, the standard is weak in the sense that it does not enforce any constraints on the implementation and complexity of the underlying algorithms. Users may be surprised by performance differences when they try different OpenMP runtime systems. In addition, OpenMP does not support distributed memory parallelization. Nonetheless, its portability, stability and maturity make it a safe long-term choice.

StarPU (*Augonnet et al., 2011*) is a runtime system that was first designed to manage heterogeneous architectures. It is a C library, which means that users are constrained to do low-level programming and use function pointers. However, it is extremely flexible and used by many HPC applications (*Agullo et al., 2010*; *Lima et al., 2019*; *Agullo et al., 2014*; *Lacoste et al., 2014*; *Agullo et al., 2013*; *Carratalá-Sáez et al., 2020*; *Sukkari et al., 2018*; *Moustafa et al., 2018*; *Bramas et al., 2020*). It also supports distributed memory parallelization through two different approaches (*Agullo et al., 2017b*). Each of these approaches uses a different description of the task graph and the amount of information that the StarPU instances have on the complete graph is different between approaches (there is one StarPU instance per computing node). The first approach is the most trivial. It consists in declaring the complete graph on all computing nodes. This means that there is one thread that describes the graph in each StarPU instance. Each instance can analyze the graph without any communication, since it holds the complete graph. This can be used to

create low-cost scheduling strategies. For example, consider that all instances iterate over the task graph and have to decide on which computing node each task gets executed. As they have a complete view of the graph, they can assign a task to a computing node while minimizing the communication and all instances take the same decision without communicating. Moreover, the instances know where the data dependencies are located because they can track them while iterating on the graph. This can be used to submit send/receive operations accordingly and manage the communication automatically. However, there is a clear disadvantage to this approach: the method cannot scale because its cost and overhead increase with the size of the task graph, independently of the number of computing nodes that will be used. In the second approach, each instance declares a partial task graph covering only the tasks it will compute. However, StarPU needs additional information to track data movement and to connect the different partial task graphs together to be able to manage the communication. One option is to request explicit communication calls (similarly to MPI) to connect the tasks between instances. Another option is for each instance to insert the tasks that are at the frontier of its partial graph. These frontier tasks are computed by the other instances. The partial graph description removes abstraction because the developers have to manually split the task graph and manage the boundaries of the partial graph. Moreover, each instance has only a partial view, making analysis and scheduling difficult. Specx uses a similar approach.

PaRSEC (*Bosilca et al., 2013*; *Danalis et al., 2014*) is a runtime system based on the parameterized task graph (PTG) model (*Cosnard & Loi, 1995*). It has been demonstrated to be effective in various scientific applications. The PTG is a domain-specific language (DSL) that captures a static, algebraic description of a task graph that can be expanded efficiently at runtime. This allows PaRSEC to manage large graphs without fully instantiating them. This approach works well on affine loops thanks to polyhedral analysis. The data-flow analysis of a task instance is constant in time and the representation of the graph is constant in space. This makes the PTG a very efficient way to represent task graphs. However, the PTG is not as expressive as other task graph models. It is difficult to use the PTG to represent applications with irregular or sparse algorithmic or data access patterns. Despite this limitation, the PTG has been shown to be effective in a wide variety of scientific applications. It is a powerful tool for parallelizing applications that can be expressed in terms of affine loops. While it is theoretically possible to write PTGs for highly dynamic applications, this would imply an unbounded amount of time spent building and traversing dynamic meta-data in memory. This is why the PTG is impractical for implementing applications with irregular or sparse algorithmic or data access patterns, where the logic is difficult to express with linear equations. The PTG graph representation is highly efficient, but the expressiveness of the model is limited. Internally, the representation allows a task graph to be collapsed into two dimensions, *i.e.*, time and parallelism (62), which enables several optimizations. The different Parsec instances all hold an algebraic representation of the complete graph in distributed memory. Parsec uses advanced mechanisms to allow efficient scheduling of the tasks using heuristics and potential input from the users.

Charm++ (*Kale & Krishnan, 1993*; *Chamberlain, Callahan & Zima, 2007*; *Acun et al., 2014*) is an object-oriented parallel programming framework that relies on a partitioned global address space (PGAS) and supports the concept of graphs of actors. It includes parallelism by design with a migratable-objects programming model and it supports task-based execution. The actors (called *chares*) interact with each other using invocations of asynchronous methods. However, with Charm++, there is no notion of tasks as we aim to have. Instead, tasks are objects that communicate *via* message exchanges. Charm++ schedules the chares on processors and provides an object migration and a load balancing mechanism. PGAS allows data to be accessed independently of its actual location, which is the inverse of what the task-based method intends to offer. A task is a piece of work that should not include any logic or communication. This approach forbids many optimizations and mechanisms that task graphs support (*Thibault, 2018*).

HPX (*Kaiser et al., 2014*; *Kaiser et al., 2020*) is an open-source implementation of the ParalleX execution model. Its implementation aims to follow the C++ standard, which is an asset for portability and compliance with existing C++ source code. In HPX, tasks request access to data by calling an accessor function (get/wait). The threads provide the parallelism description, which is tied to the order and type of data accesses.

OmpSs (*Duran et al., 2011*; *Perez, Badia & Labarta, 2008*; *Aguilar Mena et al., 2022*) uses the insert-task programming model with pragmas similar to OpenMP through the Nanos++ runtime to manage tasks. When running in distributed memory, it follows a master-slave model, which may suffer from scalability issues as the number of available resources or the problem size increases.

XKaapi (*Gautier et al., 2013*) is a runtime system that can be used with standard C++ or with specific annotations, but it requires a specific compiler. Legion (*Bauer et al., 2012*) is a data-centric programming language that allows parallelization with a task-based approach. SuperGlue (*Tillenius, 2015*) is a lightweight C++ task-based runtime system. It manages the dependencies between tasks using a data versioning pattern. X10 (*Charles et al., 2005*) is a programming model and a language that relies on PGAS. Hence, it has similar properties to Charm++. Intel Threading Building Blocks (https://www.threadingbuildingblocks.org/) (ITBB) is an industrial runtime system provided as a C++ library. It is designed for multicore parallelization or to be used in conjunction with oneAPI, but it follows a fork-join parallelization pattern.

Regarding distributed parallelization, most runtime systems can be used with MPI (*Snir et al., 1998*). The developers implement a code that alternates between calls to the runtime system and MPI communication calls. When supported by the runtime system, the data movement between CPUs/GPUs and in-node load balancing are delegated to the runtime system. More advanced methods have been elaborated and they entirely delegate the communications to the runtime system (*Zafari, Larsson & Tillenius, 2019*; *Zafari, 2018*; *Kemp & Chapman, 2018*; *Hoque & Shamis, 2018*; *Fraguela & Andrade, 2019*), like Parsec, StarPU (*Agullo et al., 2016a*), Legion, Charm++ (*Jain et al., 2015*), TaskTorrent (*Cambier, Qian & Darve, 2020*) and HPX. Some runtime systems implement the communication using MPI, others use different approaches such as libfabric (https://ofiwg.github.io/libfabric/), LCI (*Dang et al., 2018*) (used by HPX), and Gasnet (*Bonachea & Hargrove,*

*2017*) (used by UPC/UPC++ (*Zheng et al., 2014*; *Bachan et al., 2017*; *Bachan et al., 2019*), Legion and Chapel (*Wheeler et al., 2011*; *Hayashi, Paul & Sarkar, 2021*).

Most of these tools support core aspects of a task based runtime system, including the creation of a task graph (although the implementation may vary) where tasks can read or write data. However, scheduling is an important factor for performance (*Agullo et al., 2016b*) and few of these runtime systems offer a way to easily create a scheduler without having to modify the runtime system source code. Moreover, specific features allow the degree of parallelism to be increased. For instance, some runtime systems allow the user to specify that a data access is commutative, implying that tasks write data in any order. These kinds of advanced functions can significantly impact performance (*Agullo et al., 2017a*). The runtime systems differ on whether the task graphs are statically or dynamically generated, how the generation is performed, which in-memory representation is used, which parallelization levels are supported and many other features (*Thoman et al., 2018*; *Gu & Becchi, 2019*; *Gurhem & Petiton, 2020*).

We summarize the different features and specificities of some runtime systems in Table 1. We may characterize a runtime system according to four aspects: interface (how it is used), scheduling (how it distributes the workload and manages transfers and communications), features (what it can do) and overhead (how it is implemented, which may be constrained by its interface or features). According to this description and in terms of overall quality, robustness, completeness and maturity, StarPU is the most advanced runtime system. Specx offers two advantages over StarPU however: it has a C++ design/interface and therefore adheres to object-oriented programming and it supports speculative execution.

## Speculative execution

Speculative execution is an approach that can increase the degree of parallelism. It has been widely used in hardware and is an ongoing research topic in software (*Estebanez, Llanos & Gonzalez-Escribano, 2016*; *Khatamifard, Akturk & Karpuzcu, 2018*; *Martinez Caamaño et al., 2017*). The key idea of speculative execution is to utilize idle components to execute operations in advance at the risk of performing operations that may later be invalidated. The prominent approach is to parallelize an application and to detect at runtime if race conditions or an invalid access order which violates dependencies happen. The detection of invalid speculative execution can be expensive and, as a result, some research is intended to design hardware modules for assistance (*Steffan et al., 2000*; *Salamanca, Amaral & Araujo, 2018*). However, these low-level strategies are unsuitable for massively parallel applications and enforce the need for either detecting the code parts suitable for speculation or relying on explicit assistance from developers.

In a previous study, we have shown that characterizing accesses as 'maybe-write' instead of 'write' allows us to increase the degree of parallelism thanks to speculative execution in the task-based paradigm (*Bramas, 2019*). This novel kind of uncertain data access (UDA) can be used when it is uncertain at task insertion time whether the tasks will modify some data or not. Similarly to the 'commutative write', developers simply provide additional information to the runtime system, enabling it to set up a strategy by modifying the task

**Table 1  Comparison of task-based runtime systems.**

| Runtime | Interface | | Scheduling | | Features | | | | | |
| | STF | C++ | Dynamic | Custom. | CUDA | HIP | SYCL | Distributed | Execution simulation | Speculative execution |
| --- | --- | --- | --- | --- | --- | --- | --- | --- | --- | --- |
| StarPU | ✓ | ~ | ✓ | ✓ | ✓ | ✓ | ✗ | ✓ | ✓ | ✗ |
| Legion | ✗ | ✓ | ✓ | ✗ | ✓ | ~ | ✗ | ✓ | ✗ | ✗ |
| PaRSEC | ~ | ✗ | ✓ | ✗ | ✓ | ✗ | ✗ | ✓ | ✗ | ✗ |
| Chapel | ✗ | ✗ | ✓ | ✗ | ✓ | ✓ | ✗ | ✓ | ✗ | ✗ |
| Charm++ | ✗ | ✓ | ✓ | ✗ | ✓ | ~ | ✗ | ✓ | ✗ | ✗ |
| HPX | ✗ | ✓ | ✓ | ✗ | ~ | ✗ | ✗ | ✓ | ✗ | ✗ |
| OmpSs | ✓ | ~ | ✓ | ✗ | ✓ | ✗ | ✗ | ✓ | ✗ | ✗ |
| OpenMP | ✓ | ~ | ✓ | ✗ | ✓ | ~ | ~ | ✗ | ✗ | ✗ |
| Taskflow | ✗ | ✓ | ✓ | ✗ | ~ | ✗ | ✗ | ✗ | ✗ | ✗ |
| Specx | ✓ | ✓ | ✓ | ✓ | ✓ | ✓ | ✗ | ✓ | ✗ | ✓ |

**Note:**
The icon ~ means "partially supported" or "requires the user to implement part of the functionality". A runtime system supports customized schedulers if its software design and functions make it easy to create new schedulers.

graph on the fly. This also makes it possible to delay some decisions from implementation/compile time to execution time, where valuable information about the ongoing execution is available. We have implemented this mechanism in our task-based runtime system Specx (originally called SPETABARU) and conducted an evaluation on Monte Carlo simulations, which demonstrated significant speedups. We are currently developing a new model (*Souris, Bramas & Clauss, 2023*).

Furthermore, speculation has also been used in a tasking framework for adaptive speculative parallel mesh generation (*Tsolakis, Thomadakis & Chrisochoides, 2022*) and for resource allocation in parallel trajectory splicing (*Garmon, Ramakrishnaiah & Perez, 2022*).

## SPECX'S FEATURES, DESIGN AND IMPLEMENTATION

### Task graph description

In Specx, we dissociated the task graph from the so-called compute engine that contains the workers. Therefore, the user has to instantiate a task graph and select among two types, one with speculative execution capability and one without, which allows the speculative execution management overhead to be removed when no UDAs are used. We provide an example in Code 1.

**Code 1.  Specx example—creating a task graph.**

```
1  // Create a task graph
2  SpTaskGraph<SpSpeculativeModel::SP_NO_SPEC> tg;
3  // Legacy version, create a runtime (a compute engine + a task graph)
4  SpRuntime runtime(SpUtils::DefaultNumThreads());
```

**Task insertion:** Specx follows the STF model: a single thread inserts the task in the runtime system (task graph object) and specifies which variables will be written to or read from. Additionally, the user can pass a priority that the scheduler is free to use when making decisions. The core part of the task consists in a callable object with the operator *()*, which allows the use of C++ lambda functions. The data access modes that Specx currently supports are:

- SpRead: the given dependency will only be read by the task. As such, the parameter given to the task callable must be *const*.
- SpWrite: the given dependency will be written to or read and written to by the task.
- SpCommutativeWrite: the given dependency will be written to or read and written to by the task but the execution order of all the jointly inserted *SpCommutativeWrite*s is not important.
- SpMaybeWrite: the given dependency might be written to or read and written to by the task. Possible speculative execution patterns can be applied.
- SpAtomicWrite: the given dependency will be written to or read and written to by the task but the user will protect the access *via* its own mechanisms (using mutual exclusion for example). The runtime system manages this access very similarly to a read access (multiple *SpAtomicWrite*s can be done concurrently but the runtime system has to take care of the read-after-write, write-after-read coherency).

When a dependency *X* is passed, the runtime dereferences *X* to get its address and this is what will be used as the dependency. An important point when using task-based programming is that it is the user's responsibility to ensure that the objects are not destroyed before all tasks that use them are completed. We provide an example in Code 2.

---

**Code 2.** **Specx example—creating a task for the CPU.**

```
1 const int initVal = 1;
2 int writeVal = 0;
3 // Create a task with lambda function
4 tg.task(SpRead(initVal), SpWrite(writeVal),
5             [](const int& initValParam, int& writeValParam){
6        writeValParam += initValParam;
7 });
```

---

**Dependencies on a subset of objects:** A critical drawback of OpenMP 4 was the rigidity of dependency declaration. Indeed, the number of dependencies of a task had to be set at compile time. Since OpenMP 5, it is now possible to declare an iterator to express dependencies on several addresses at once. This is extremely useful if, for example, we want to declare a dependency on all the elements of a vector.

To solve this issue, in Specx, we can declare the dependencies on a set of objects using the following constructs: *SpReadArray(<XTy> x,<ViewTy> view), SpWriteArray(<XTy> x,*

*<ViewTy> view), SpMaybeWriteArray(<XTy> x,<ViewTy> view), SpCommutativeWriteArray (<XTy> x,<ViewTy> view), SpAtomicWriteArray(<XTy> x,<ViewTy> view)*, where *x* should be a pointer to a contiguous buffer (or any container that support the *[]* operator) and *view* should be an object representing the collection of indices of the container elements that are affected by the dependency. *view* should be iterable (in the sense of "stl iterable").

With this mechanism, Specx can iterate over the elements and apply the dependencies on the ones selected. We provide an example in Code 3.

**Code 3.** Specx example—using an array of dependencies.

```
1 std::vector<int> vec = ...;
2 // Access all the elements in the SpArrayView
3 tg.task(SpPriority(1), SpWriteArray(vec.data(),SpArrayView(vec.
  size()))),
4            [](SpArrayAccessor<int>& vecView){
5     ...
6 });
```

**Task viewer:** Inserting a task in the task graph returns a task view object which gives access to some attributes of the complete inner task object. For instance, it allows the user to set the name of the task, to wait for task completion or to get the value produced by the task (in case the task returns a value). Unfortunately, there is a pitfall with the current design, which is the fact that accessing the task through the viewer can potentially be done after the task has been computed. For instance, we cannot use the tasks' names in the scheduler because they might be set after the tasks were computed. We provide an example in Code 4.

**Code 4.** Specx example—task viewer.

```
1 auto taskViewer = runtime.task(SpRead(initVal), SpWrite(writeVal),
2            [](const int& initValParam, int& writeValParam) -> bool {
3     writeValParam += initValParam;
4     return true;
5 });
6 taskViewer.setName("The name of the task");
7 taskViewer.wait(); // Wait for this single task
8 taskViewer.getValue(); // Get the value (when the task is over)
```

## Teams of workers and compute engines

Within Specx, a team of workers constitutes a collection of workers that can be assigned to compute engines. In the current implementation, each worker is associated with a CPU thread that continuously retrieves tasks from the scheduler and processes them. If the worker is CPU-based, the task is directly executed by the CPU thread. Conversely, in the

case of a GPU worker, the CPU thread manages the data movement between memory nodes and calls the device kernel.

A compute engine necessitates a team of workers and may be responsible for several task graphs. Currently, it is not possible to change the compute engine assigned to a task graph, but it is possible to shift workers among different compute engines. This feature provides the ability to dynamically adjust the capabilities of the compute engine during execution and design advanced strategies to adapt to the workload of the graphs.

Given that dependencies among task graphs are not shared, the insertion of tasks and their dependencies into one task graph does not affect other task graphs. This allows the creation of recursive parallelism, in which a task graph is created within a task. Such a task graph could potentially be attached to the same compute engine as the parent task. This approach could help mitigate the overhead associated with the creation of a single large set of tasks by organizing them into sub-task-graphs. We provide an example in Code 5.

**Code 5.** Specx example—creating a compute engine.

```
1 SpTaskGraph<SpSpeculativeModel::SP_NO_SPEC> tg;
2 // Create the compute engine
3 SpComputeEngine ce(SpWorkerTeamBuilder::TeamOfCpuWorkers
  (NbThreads));
4 // OR
5 SpComputeEngine ce(SpWorkerTeamBuilder::TeamOfCpuCudaWorkers());
6 // Tells which compute engine will execute the graph
7 tg.computeOn(ce);
```

## Tasks for heterogeneous hardware

Specx follows the same principles as StarPU to support heterogeneous hardware, *i.e.*, we have distinct workers for each type of processing unit and each task can be executed on CPUs, GPUs or both. Specifically, at task insertion, we require a unique callable object for each processing unit type capable of executing the task. During execution, the scheduler determines where the task will be executed. This represents a critical challenge in task-based computing on heterogeneous systems.

Regarding the interface, the primary challenge is the data movement between memory nodes. More specifically, we strive to exploit C++ and use an abstraction mechanism to facilitate object movement. Consequently, we have determined that objects passed to tasks should comply with one of the following rules: (1) the object is trivially copyable (https://en.cppreference.com/w/cpp/types/is_trivially_copyable); (2) the object is an std::vector of trivially copyable objects; (3) the object's class implements specific methods that the runtime system will call.

In the last case, the object's class must have a class attribute of a data type called *DataDescriptor* and three methods:

- *memmovNeededSize*: Invoking this method on the object should yield the required memory size to be allocated for the object copy on the device.
- *memmovHostToDevice*: This method is called to transfer the object to the device. The method receives a mover class (with a copy-to-device method) and the address of a memory block of size determined by *memmovNeededSize* as parameters. The method may return a *DataDescriptor* object, which will later be passed to *memmovDeviceToHost* and to the task using the object.
- *memmovDeviceToHost*: This method is invoked to move the data from the GPU back to the object. The method receives a mover class (with a copy-from-device method), the address of a GPU memory block and an optional *DataDescriptor* object as parameters.

From a programming perspective, we require the users to determine how the data should be moved as they have the knowledge to do so. For example, consider an object on a CPU being a binary tree where each node is in a separate memory block. It would be inefficient to allocate and copy each node. Consequently, we ask the users to estimate the memory block size needed and we perform a single allocation. Then it is the users' responsibility to mirror the tree on the GPU using the block we allocated, and to implement the task such that it can use the mirrored version. This design may change in the future as we continue to apply Specx to existing applications.

Currently, we employ the Least Recently Used (LRU) policy to determine which memory blocks should be evicted from the devices when they are full. Concretely, this implies that when a task is about to be computed on the device, the worker's thread will iterate over the dependencies and copy them onto the GPU's memory using a stream/queue. If an object already has an up-to-date version on the device, the copy will be skipped and if there is not enough free memory, older blocks will be evicted. As a result, at the end of an execution, the up-to-date versions of the objects might be on the GPUs, necessitating a transfer back to the CPUs. At present, this can be accomplished by inserting empty CPU tasks that use these objects.

By default, worker teams align with hardware configurations, *i.e.*, they will contain GPU workers for each available GPU. Therefore, if users are not careful and only use one type of processing unit for their tasks, the hardware will be underutilized as some workers will remain idle.

We provide an example of a task that can be executed on both the CPU and the GPU and a class that provides a copy method in Code 6.

**Code 6.** Specx example—creating a task for both the CPU and the GPU.

```
1  class Matrix{
2      int nbRows;
3      int nbCols;
4      std::vector<double> values;
5  public:
```

```
6        // What to allocate on the device
7      std::size_t memmovNeededSize() const{
8            return sizeof(double)*nbRows*nbCols;
9      }
10
11     // Copy to the device (size == memmovNeededSize())
12     template <class DeviceMemmov>
13     auto memmovHostToDevice(DeviceMemmov& mover, void* devicePtr,
       std::size_t size){
14           double* doubleDevicePtr = reinterpret_cast<double*>
             (devicePtr);
15           mover.copyHostToDevice(doubleDevicePtr, values.data
             (),... nbRows*nbCols*sizeof(double));
16           return DataDescr{rowOffset, colOffset, nbRows, nbCols};
17     }
18
19     // Copy to the CPU
20     template <class DeviceMemmov>
21     void memmovDeviceToHost(DeviceMemmov& mover, void* devicePtr,
       std::size_t size, const... DataDescr& /*inDataDescr*/){
22           double* doubleDevicePtr = reinterpret_cast<double*>
             (devicePtr);
23           mover.copyDeviceToHost(values.data(),
             doubleDevicePtr,... nbRows*nbCols*sizeof(double));
24     }
25 };
26
27 // ....
28 Matrix matrix;
29
30 tg.task(SpPriority(1), SpWrite(matrix),
31     SpCpu([](Matrix& matrix){
32         // ...
33       })
34 #ifdef SPECX_COMPILE_WITH_CUDA
35     , SpCuda([](SpDeviceDataView<Matrix> matrix) {
36           // ...
37       })
38 #endif
39 ).setTaskName(std::string("My operation")); // Set the name of the
   task
```

In most GPU programming models, a CPU thread must explicitly request the use of one of the GPUs available on a machine (*e.g.*, using `cudaSetDevice`). Once a device is selected, all subsequent calls to the GPU are executed on that specific device. However, selecting a device incurs overhead due to context creation. When a CPU thread needs to compute a task that requires updated data stored in the memory of one or several GPUs, it must first copy the data to the system's RAM. To address this, we have implemented a background CPU thread for each GPU, which can interact with the GPU. When a CPU worker needs to interact with a GPU, it does so through the background thread using a spawn/sync mechanism.

## Mixing communication and tasks

In the context of distributed memory parallelization, Specx provides the capability to mix send/receive operations (MPI) and computational tasks. Putting MPI communications directly inside tasks will fail due to the potential concurrent accesses to the communication library (which is not universally supported by MPI libraries) and the risk of having workers wait inside tasks for communication completion, leading to deadlocks if tasks sending data on one node do not coincide with tasks receiving data on another. Therefore, to avoid having the workers deal with communication, our solution is to use a dedicated background thread that manages all the MPI calls.

In this approach, a *send* operation is transformed into a communication task that does a read access on the data. It is executed by the background thread. Similarly, a *receive* operation becomes a communication task that performs a write access and is also processed by the background thread. Once a communication task is ready, the background thread executes the corresponding non-blocking MPI calls, receiving an MPI request in return. This request is stored in a list. The background thread tests the requests in the list for completion by calling the MPI *Testany* function. When a request is fulfilled, the background thread releases the dependencies of the associated communication task, thereby ensuring task graph execution progresses as soon as possible.

In order to send/receive C++ objects using MPI in a single communication (although we perform two—one for the size and one for the data), we need a way to store the object into a single array. To achieve this, the object must comply with one of the following rules:

- It should be trivially copyable;
- It should provide access to a pointer to the array to be sent (or received). For example, if a class has virtual methods, it will not be trivially copyable. However, if the class's only attribute is a vector of integers, sending the object is equivalent to sending the vector's data.
- It should support our serialization/deserialization methods. Here, we allow the object to serialize itself using our utility serializer class, yielding a single array suitable for communication. Upon receipt, the buffer can be deserialized to recreate the object. This method offers the most flexibility, but it is also the least efficient.

Specx also supports MPI broadcast as part of the MPI global communication functions. Currently, users must ensure that all Specx instances perform the same broadcasts and in the same order.

As a side note, MPI communications are incompatible with the speculative execution capabilities of Specx due to diverse execution paths and additional tasks being potentially instantiated.

We provide an example of a distributed implementation in Code 7.

**Code 7.** Specx example—sending/receiving a matrix object (using the serializer method).

```
1   class Matrix{
2     // ...
3     Matrix(SpDeserializer &deserializer)
4         : nbRows(deserializer.restore<decltype(nbRows)>("nbRows")),
5           nbCols(deserializer.restore<decltype(nbCols)>("nbCols")),
6           values(deserializer.restore<decltype(values)>("values")){
7     }
8
9     void serialize(SpSerializer &serializer) const {
10        serializer.append(nbRows, "nbRows");
11        serializer.append(nbCols, "nbCols");
12        serializer.append(values, "values");
13     }
14  };
15
16  // ....
17  tg.mpiRecv(matrix, srcRank, tag);
18  // ....
19  tg.mpiSend(matrix, destRank, tag);
```

## Scheduling

While we designed the scheduler module following the implementation approach used in StarPU, we reimplemented everything, especially because StarPU is written in C and Specx is written in C++.

A scheduler provides two key functions: *push* and *pop*. When a task becomes ready (*i.e.*, its predecessors have finished), it is pushed into the scheduler. Conversely, when a worker becomes available, it calls the pop function on the scheduler, which may potentially return `null` if there are no tasks compatible with its processing unit type or if the scheduler determines it is not appropriate to assign a task. Thus, the scheduler plays a crucial role in the management of task distribution and the order of execution of tasks.

At present, Specx includes several simple schedulers. The first one is a LIFO (Last In, First Out) scheduler, called `SpSimpleScheduler`, which uses a singly linked list implemented with atomic operations. This scheduler only supports CPU workers.

The second scheduler, `SpPrioScheduler`, uses a priority queue to store tasks, ensuring they are popped in priority order. This scheduler also only supports CPU workers.

The third scheduler, called `SpHeterogeneousPrioScheduler`, uses three priority queues: one for CPU-only tasks, one for GPU-only tasks and one for hybrid tasks, *i.e.*, tasks that can be executed by CPUs as well as GPUs. Workers first pick a task from their corresponding queue and then from the hybrid queue if necessary. Consequently, tasks within a queue are consumed in priority order but tasks of higher priority might reside in the hybrid queue.

Finally, we provide a variant of the heteroprio scheduler (*Agullo et al., 2016b*; *Flint, Paillat & Bramas, 2022*), called `SpMultiPrioScheduler`. In our implementation, each GPU worker has its own queue. The user should provide a priority-pair for each task, where the pair represents a priority for the CPU and a priority for the GPU. Thus, tasks are inserted into the queue with different priorities for the CPU and the GPU (the idea being that a task for which the GPU is faster should have a higher priority on the GPU, and vice versa for the CPU). This scheduler includes an option to favor the worker that releases the task by providing a bonus to the task's priorities.

We plan to introduce more sophisticated schedulers (*Tayeb et al., 2024*) in the near future, but this will require performance models of the tasks to enable the scheduler to make more informed decisions, which implies significant engineering work.

In any case, with our design, it is straightforward to provide a new scheduler, and users have the flexibility to implement a custom scheduler specifically designed for their application. This can be accomplished by creating a new class that inherits from our abstract scheduler interface.

## Speculative execution

Specx supports task-based speculative execution (*Bramas, 2019*), which is an ongoing research problem. We currently support two speculative models applicable when certain data accesses are flagged as *maybe-write*. In these scenarios, the runtime system may duplicate some data objects and tasks to enable potential speculative work, subsequently performing a rollback if the uncertain tasks actually modified the data. The present article leaves the presentation and study of speculative execution aside.

## Internal implementation

In this section, we delve into the finer details of Specx's implementation. When a task is inserted, the callable's prototype should match the dependency types. Hence, read parameters are passed as *const*. For CPU callables, parameters should be of type references to the object types passed as arguments. Using a value instead of a reference will simply result in a copy, which is typically not the intended outcome. Indeed, if values are required but are not significant as dependencies, it is more appropriate to pass them as captures in a lambda function.

When an object is passed to a task, the runtime system dereferences it to obtain its address. This address is utilized as a dependency value and also as a key in an unordered hashmap that matches pointers to data handles. A data handle is a class that contains all the necessary information the runtime system requires concerning a dependency. For instance, the data handle includes the list of dependencies applied to the associated object. Progress through the list ensues when dependencies get released. In terms of implementation, we do not construct a graph; instead, we have one data handle per address that has been used as a dependency and the data handle's dependency list contains pointers to the tasks that use the related objects. Consequently, when a task is finished, we increment a counter on the dependency list and access the next tasks. Doing so, we then examine whether the task now pointed to by the updated counter is ready and we push the task into the scheduler if that is the case. The data handle also has a mutual exclusion object that enables its locking for modification. When several data handles need to be locked, we sort them based on their address, ensuring deadlock prevention.

The commutative dependency (*SpCommutativeWrite*) is managed differently because the related tasks' order is not static. Said differently, when the next tasks use the commutative access, we do not know which one should be executed. As such, we cannot merely point to the first task in the dependency list and stop our inspection if it's not ready, as the following tasks that also have commutative access to the dependency might be ready. This necessitates a check on all the tasks performing a commutative access at the same point in time. However, several threads that completed a task might do so simultaneously, requiring us to use a mutual exclusion to protect all the commutative dependencies. In other words, using commutative dependencies implies the use of global mutual exclusion.

We use C++ meta-programming massively, such as to test if an object is trivially copyable or supports serialization methods for example. We also use the inheritance/interface and the template method design pattern. For instance, this allows us to have a task class containing the callable type and thus the types of all parameters and arguments. We can then carry out meta-programming tests on the arguments to ensure compliance with specific rules, *etc.*

Finally, since we use a hashmap to store the data dependency objects' information, using their address as keys, it is currently undefined behavior to have objects of different types stored at the same address. This primarily occurs when an object of type $x$ is freed and, subsequently, an object of type $y$ is allocated using the same memory block. This is because the data handle class uses a data copier through an interface, but the copier is actually templatized over the dependency object type.

### Visualization

Profiling and optimizing task-based applications is crucial to achieve high performance. The main informations are:

- Degree of parallelism: This represents how many tasks can be executed in parallel. The task graph can be used to evaluate if the degree of parallelism is sufficient to fully utilize

**Peer**J Computer Science

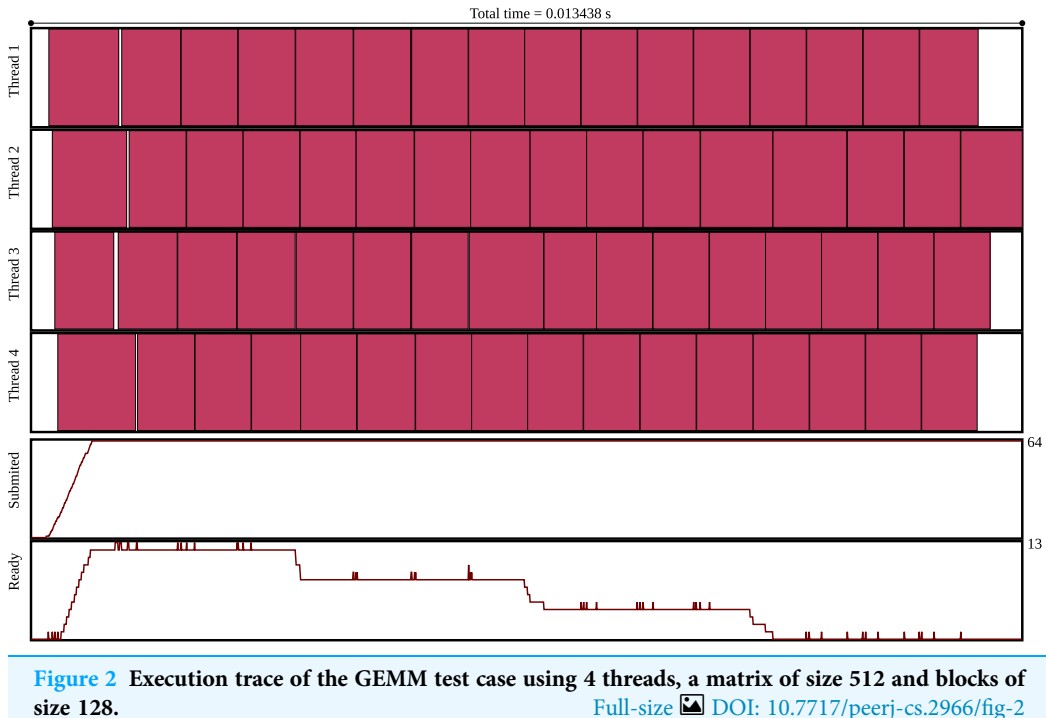

**Figure 2 Execution trace of the GEMM test case using 4 threads, a matrix of size 512 and blocks of size 128.**

all the processing units. Furthermore, the number of ready tasks over time during the execution can also be analyzed.

- Task granularity: The task granularity can impact the degree of parallelism. An execution trace analysis can help determine if the granularity is too small. If so, the overhead of task management and/or data displacement may be too large compared to the duration of the actual computation performed by the task, thereby negatively affecting performance. Conversely, if the tasks are too large, the degree of parallelism can be too small and the end of the execution can be penalized when there are too few tasks to compute.

- Scheduling choices for task distribution: If a task is assigned to a type of worker that can compute the task but is not as efficient at it as other types of workers, it can reduce performance. It could be faster to wait for more efficient workers and assign them the task, but this depends on the scheduler.

- Scheduling choices for the ordering of tasks: The degree of parallelism (and sometimes the availability of tasks for each worker type) can be influenced by the task execution order, that is, the choice among the tasks that are ready to be computed.

A good situation, but not necessarily optimal, is when no worker has been idle and the tasks have been assigned to the processing units that can execute them most efficiently.

To facilitate profiling, Specx provides features to export the task graph and the execution trace. The task graph is generated in the *dot* format (https://gitlab.com/graphviz/graphviz). For the execution trace, a scalable vector graphics (SVG) file (https://www.w3.org/Graphics/SVG/) (that can be opened with any modern internet browser) is exported. The execution trace also indicates the number of tasks ready for computation over time

during the execution and the number of tasks submitted over time during the execution. Specx also allows the amount of data transferred between the CPUs and the GPUs to be exported. In the next release, we plan to export metrics that will provide concise but meaningful numbers on execution quality, such as the idle time.

An execution trace is provided in Fig. 2.

---

**Code 8.** Specx example—exporting the task graph and the execution trace.

```
1  // Export the graph
2  tg.generateDot("/tmp/graph.dot");
3  // Export the execution trace (false means: hide the dependencies)
4  tg.generateTrace("/tmp/gemm-simu.svg", false);
```

---

## PERFORMANCE AND USABILITY STUDY

### Overhead estimate

In this section, we aim to estimate the overhead of using the Specx runtime system.

**Configuration:** We assess our method on the following configuration:

- Intel-AVX512: it is a $2 \times 18$-core Cascade Lake Intel Xeon Gold 6240 at 2.6 GHz with AVX-512 (Advanced Vector 512-bit, Foundation, Conflict Detection, Byte and Word, Doubleword and Quadword Instructions, and Vector Length). The main memory consists of 190 GB DRAM memory arranged in two NUMA nodes. Each CPU has 18 cores with 32 KB private L1 cache, 1,024 KB private L2 cache, and 25 MB shared L3 cache. We use the GNU compiler 11.2.0 and the MKL 2022.0.2.

**Results:** In this section, we discuss the engine overhead that we evaluate with the following pattern. We create a runtime system with $T$ CPU workers and $T$ distinct data objects. Then, we insert $T \times N$ tasks, with each task accessing one of the data objects. Consequently, the task graph we generate is actually composed of $T$ independent sub-graphs. Inside each task, the worker that executes it, simply waits for a given duration $D$. As a result, the final execution time is given by $N \times (D + O)$, where $O$ is the overhead of the runtime picking a task to execute. Also, we can measure the time it takes to insert the $T \times N$ tasks to obtain an insertion cost $I$.

We provide the result in Fig. 3 for 1 to 20 dependencies. As expected, the overheads ($O$ and $I$) are significantly higher when commutative writes are used than when normal writes are used. The insertion cost is also higher when the task duration is smaller ($D = 10^{-5}$). The reason is that when the tasks are smaller, the workers query the runtime system more often, which can compete with the insertion of ready tasks by the master thread and create contention. The insertion cost also increases slightly as the number of dependencies of the tasks increases. Finally, for the write access, the overhead per task remains stable as the number of dependencies per task increases. For the commutative access, however, both overheads ($O$ and $I$) increase with the number of dependencies.

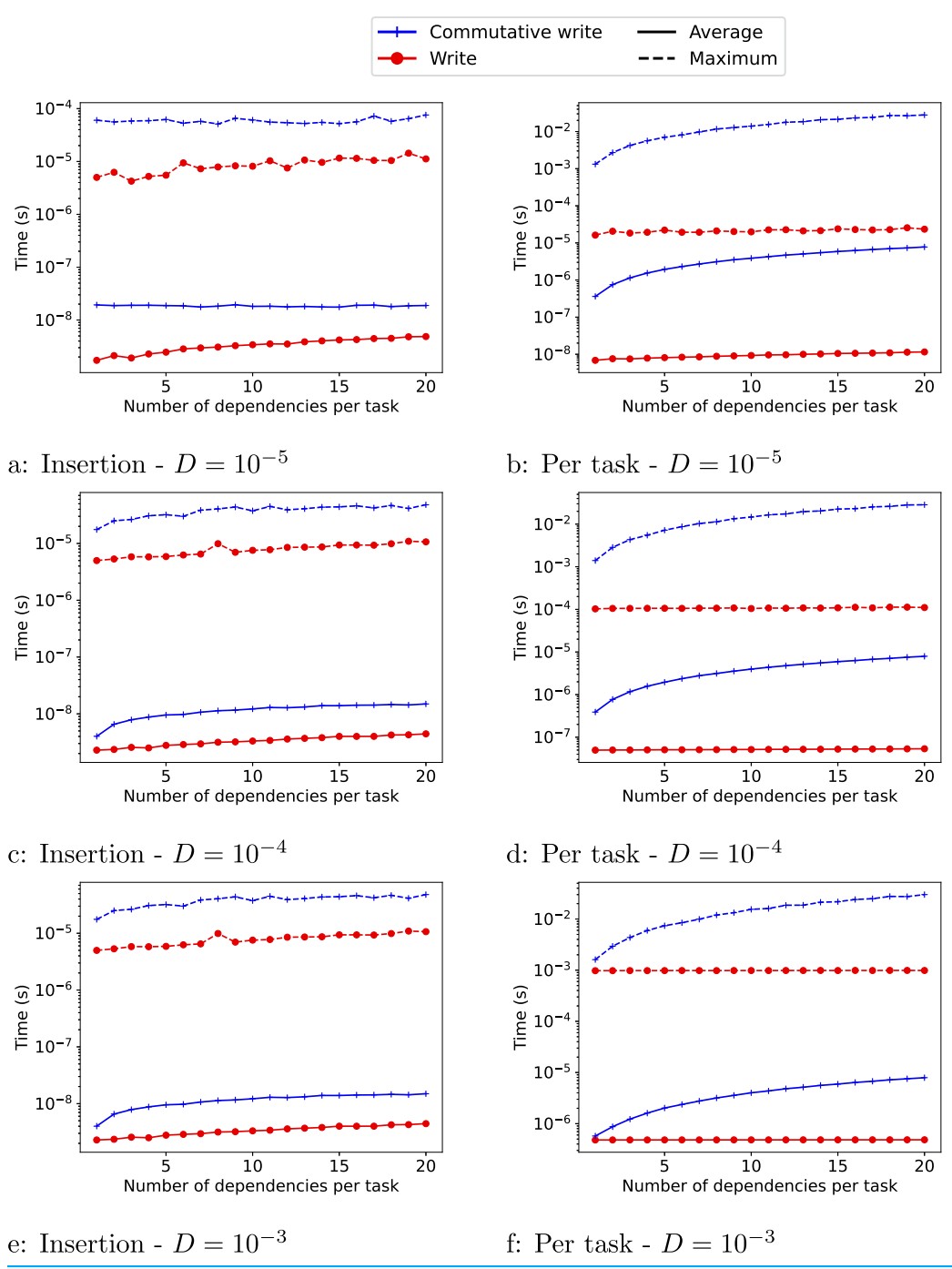

**Figure 3** **(A–F) Estimate of the overheads for the** *write* **(•) and** *commutative-write* **(+) data accesses for different numbers of dependencies.** We provide the maximum overhead reached (−−) and the average one (−). The insertion cost *I* is shown in the left column and the overhead *O* of picking a task to execute is shown in the right column.

## Numerical applications

In this section, we evaluate the performance of Specx on four numerical applications.

**Test cases:**

- Vector-scalar product (Blas axpy): In this application, we compute the axpy operation of a vector that is split into pieces. We vary the size of these pieces (the number of values computed in each task) and the number of pieces (the number of tasks). All tasks have the same computational cost and are independent. Each memory block is only used once, either for reading or writing. The kernels are implemented using our own code. We use simple floating-point precision.

- General matrix-matrix product (Blas Gemm): In this application, we split three square matrices of the same dimension into blocks/panels of the same size. We vary both the dimension of the matrices and the size of the blocks. Consequently, all tasks have the same computational cost and there only are write dependencies on the blocks of the output matrix (usually referred to as *C*). The tasks call Blas functions (Intel MKL on the CPU and cuBlas on the GPU). We use double floating-point precision.

- Cholesky factorization: In this application, we split a matrix into blocks/panels and perform a Cholesky factorization by calling Blas/Lapack functions, which results in four types of tasks (POTRF, TRSM, SYRK, and GEMM). All tasks of the same type have the same computational cost. Each task has predecessors and/or successors and some tasks are highly critical, *i.e.*, they release dependencies for a lot of tasks or are on the critical path. The tasks call Blas/Lapack functions (Intel MKL on the CPU and cuBlas on the GPU). We use double floating-point precision.

- Particle interactions (n-body): In this application, we generate groups of particles and perform two types of computations: inner (interactions of particles within the same group) and outer (interactions between groups of particles) computations. We vary the number of blocks and the number of particles in each group. As a result, most of the tasks have different computational costs. In our implementation, dependencies are applied to complete groups; we do not split the groups into symbolic data and computational data. The kernel is the computation of a direct summation or coulomb potential, giving around 17 Flops and one SQRT per interaction. We use double floating-point precision.

We recall that in our study, we are not interested in the performance of the kernels, but in the performance of the runtime system. We want to evaluate the overhead of the runtime system, the performance of the task-based approach, the scheduling choices and the data management.

**Configurations:** We use the following software configuration: NVCC 12.3, the GNU compiler 10.2.0 and Intel MKL 2020.

We tested on four different hardware platforms:

- **rtx8000:** Two NVIDIA Quadro RTX8000 GPUs (48 GB) and 2x 20-core Cascade Lake Intel Xeon Gold 5218R CPU at 2.10 GHz (190 GB). SM 7.5 (Turing architecture)

- **v100:** Two NVIDIA V100 GPUs (16 GB) and 2x 16-core Skylake Intel Xeon Gold 6142 at 2.6 GHz (380 GB). SM 7.0 (Volta architecture)
- **p100:** Two NVIDIA P100 GPUs (16GB) and 2x 16-core Broadwell Intel Xeon E5-2683 v4 at 2.1 GHz (128 GB). SM 6.0 (Pascal architecture)
- **a100:** Two NVIDIA A100 GPUs (40 GB) and 2x 24-core AMD Zen2 EPYC 7402 at 2.80 GHz (512 GB). SM 8.0 (Ampere architecture)

For each GPU we use four streams, which means that we create four CPU threads with each thread having a stream and the four threads sharing the same GPU.

### Results

We present the results for the GEMM in Fig. 4, for the Cholesky test case in Fig. 5, for the axpy test case in Fig. 6 and for the particle interaction test case in Fig. 7.

The results indicate that the performance of the GEMM test cases is significantly influenced by the block and matrix size. GPUs show a substantial speedup for matrices with dimensions of 16,384, particularly with block sizes larger than 128. This demonstrates that when GPUs do not have enough work to process within a task, the overhead associated with task management can become significant, sometimes leading to performance lower than that on the CPU. This effect is even more pronounced in the AXPY test case, where the CPU outperforms the GPU for most configurations.

Another notable effect is the poor execution performance near the end for certain configurations. For instance, in the GEMM test on the P100 GPU (Fig. 4B), the CPU experiences a performance drop for the 8,192/512 configuration. In this scenario, the workload per task is substantial and when only one or a few threads remain active to compute the final tasks, the rest of the threads remain idle, leading to a dramatic decline in speedup.

The Cholesky test cases exhibit highly variable performance. When using GPUs, the performance appears more stable, though the speedup is not as significant as in the GEMM test case. For CPU-only configurations, performance is highly inconsistent, with notable drops for block sizes of 128, among others. These variations are attributable to scheduling, which does not take into account criticality. Consequently, having a good execution order is based on luck and the performance can be highly variable.

In the particle simulations, GPUs provide a significant speedup. However, the benefit of using two GPUs instead of one is negligible. For example, on the RTX GPUs (Fig. 7A), the speedup is approximately 53 for one GPU and 60 for two GPUs when compared to sequential execution.

In Fig. 8, we present the amount of memory transfers for the GEMM test case across various matrix sizes, block sizes and schedulers. The data shows that the number of transfers consistently increases when using two GPUs compared to one GPU and the volume of transfers grows with matrix size, as expected.

Additionally, the number of transfers slightly increases with block size. This is because larger block sizes make GPUs relatively more efficient than CPUs. As a result, GPUs perform more computations and require more data to be transferred. Conversely, for small

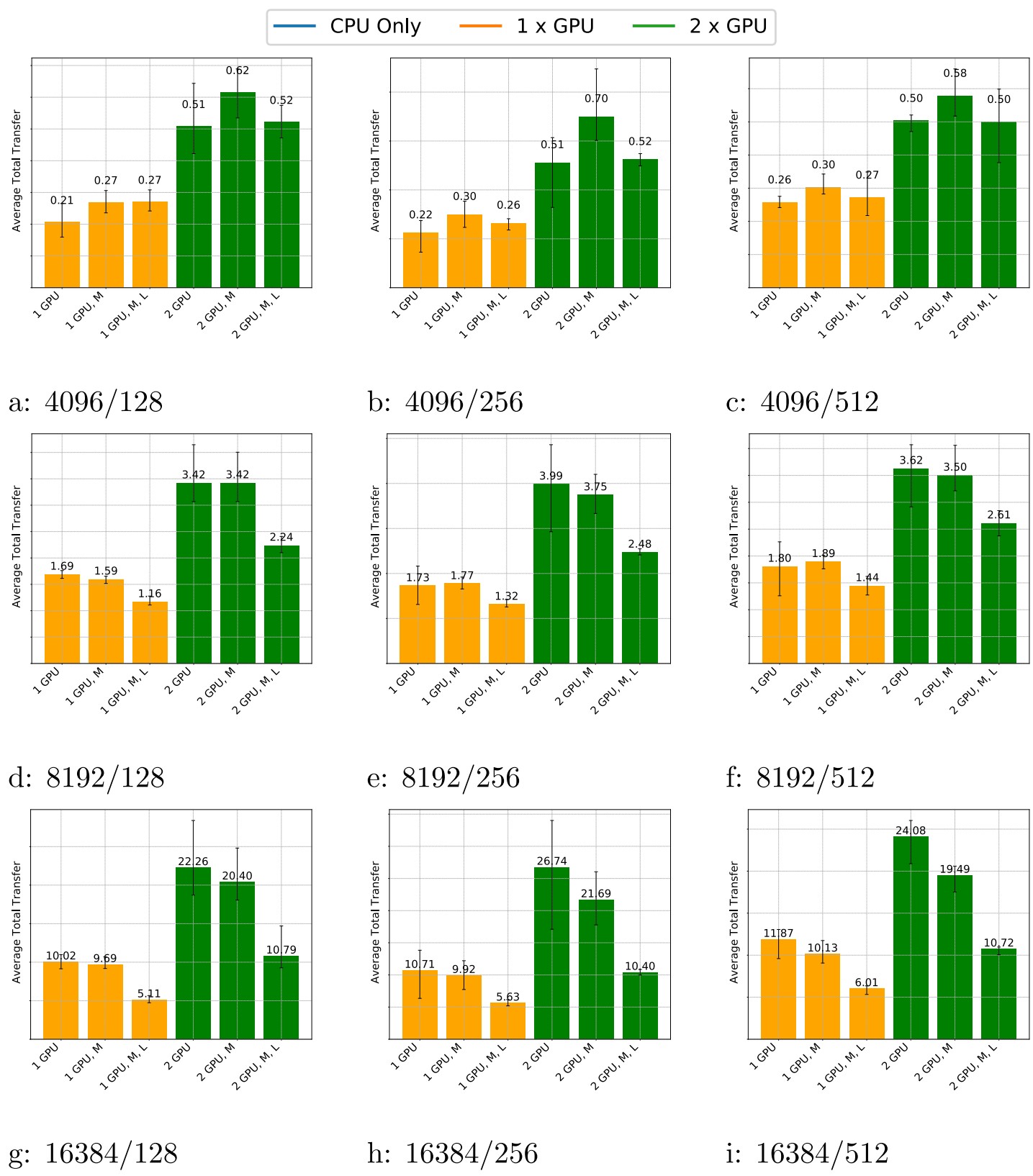

**Figure 4 (A–I) Performance results for the GEMM test case.** The *x*-axis represents the test case's size, and the *y*-axis represents the speedup over a sequential execution. The FLOPS (flops/s) of the best configuration are displayed above the plots.

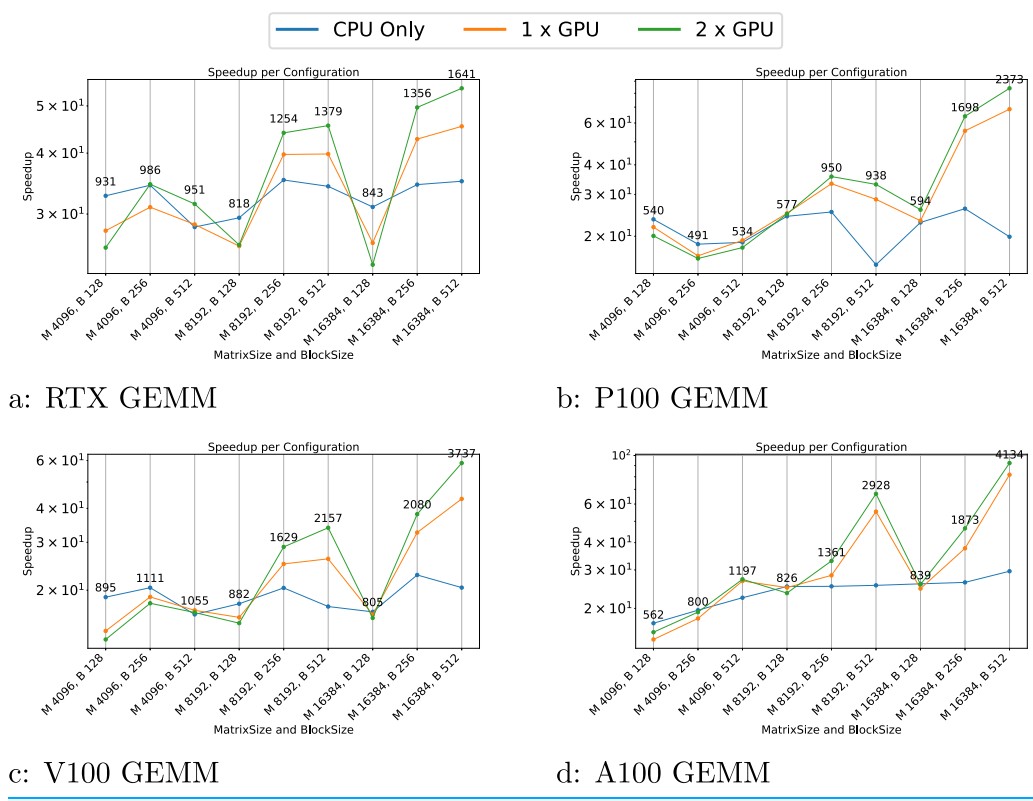

a: RTX GEMM

b: P100 GEMM

c: V100 GEMM

d: A100 GEMM

**Figure 5** **(A–D) Performance results for the Cholesky test case.** The *x*-axis represents the test case's size, and the *y*-axis represents the speedup over a sequential execution.

matrices, the performance benefit of GPUs is less pronounced; they compute fewer tasks and the advantages of using the multiprio scheduler or the locality feature are diminished.

For large matrices, the multiprio scheduler generally provides better performance and the locality feature is consistently beneficial. This improvement is due to the multiprio scheduler with the locality feature, which ensures that each worker computes on the same *C*-panel. Although this approach is not fully optimal, it reduces the number of transfers and enables potential reuse of *A* or *B*-panel data.

All in all, the results show that the performance of Specx is highly dependent on the application and the hardware configuration. The runtime system is capable of providing significant speedups, particularly when using GPUs, but the performance can be highly variable. The scheduling choices can have a significant impact on the performance and the data management can also be crucial.

### Comparison with OpenMP and StarPU

In Fig. 9, we compare Specx against two implementations, one using OpenMP and one using StarPU. For StarPU, we also used two different schedulers. The first, *lws*, is a very naive scheduler in which each worker has a queue of ready tasks. When a worker's queue is empty, the worker steals a task from another queue. Additionally, when a worker releases tasks, the scheduler stores them in that worker's queue. The second scheduler is *DMDA*, a

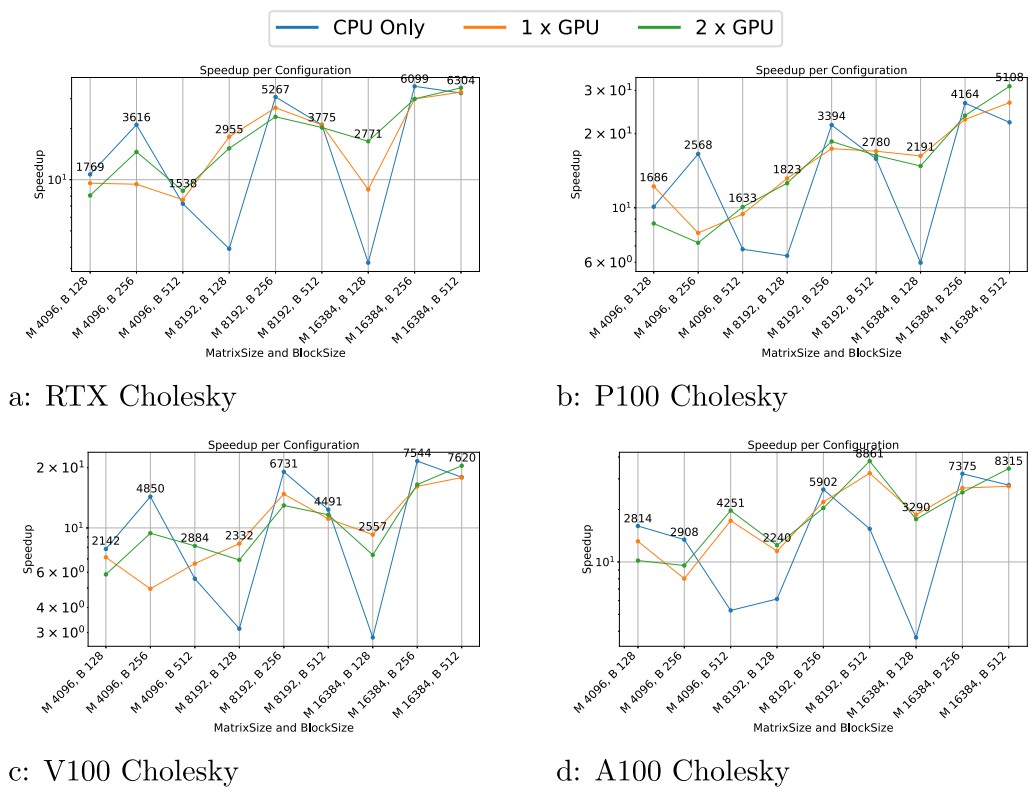

**Figure 6** **(A–D) Performance results for the axpy test case.** The *x*-axis represents the test case's size, and the *y*-axis represents the speedup over a sequential execution. The FLOPS (flops/s) of the best configuration are displayed above the plots.

very powerful and efficient scheduler. To leverage the full potential of this scheduler, one would have to build performance models of the tasks, which we did not do however, because we wanted to keep the implementation model-free.

The execution time of the OpenMP implementation is very low, even compared to the *StarPU-LWS* implementation. Both OpenMP and *StarPU-LWS* execute tasks naively without attempting to reduce memory transfers, but StarPU might reuse data moved to the GPUs for several tasks (this is different from making scheduling decisions based on data locality which *lws* does not), which is not the case for our OpenMP implementation.

From these comparisons, we observe that Specx offers slightly better performance than the other runtimes in this test case with these implementations. However, we can't claim that Specx is superior to other runtime systems. OpenMP is portable and has low overhead. Furthermore, one could implement the particle simulation test case differently—especially if one implements a mechanism to reuse data on the GPU—but that would require building functionality that is performed transparently by the runtime itself in the case of Specx or StarPU. In addition, StarPU offers many more features than Specx and could deliver better performance in many applications thanks to its performance models and advanced schedulers.

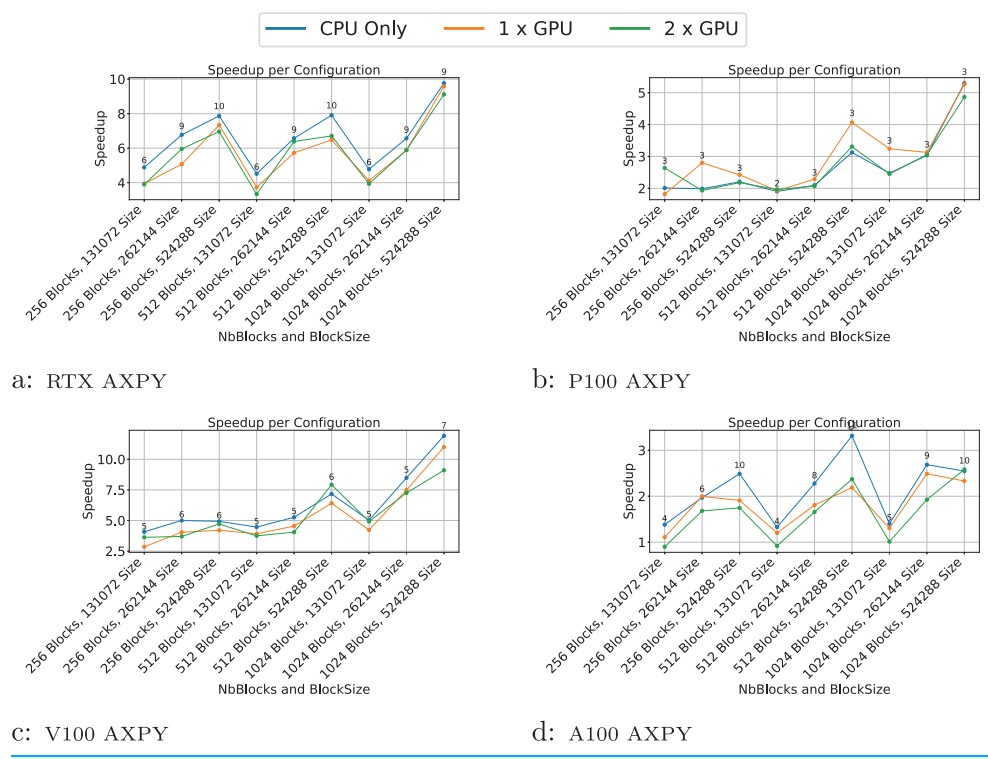

**Figure 7 (A–D) Performance results for the particle test case.** The *x*-axis represents the test case's size, and the *y*-axis represents the speedup over a sequential execution.

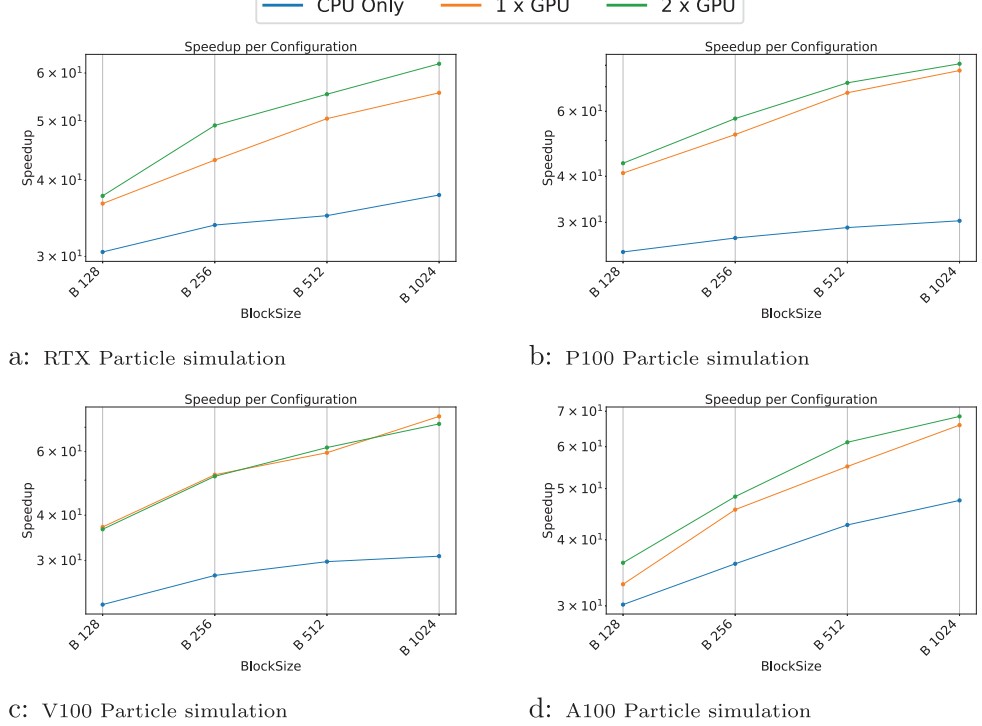

**Figure 8 (A–D) Transfers for the GEMM test case with different matrix and block sizes and different schedulers (*M* for multiprio, *L* for locality feature).**

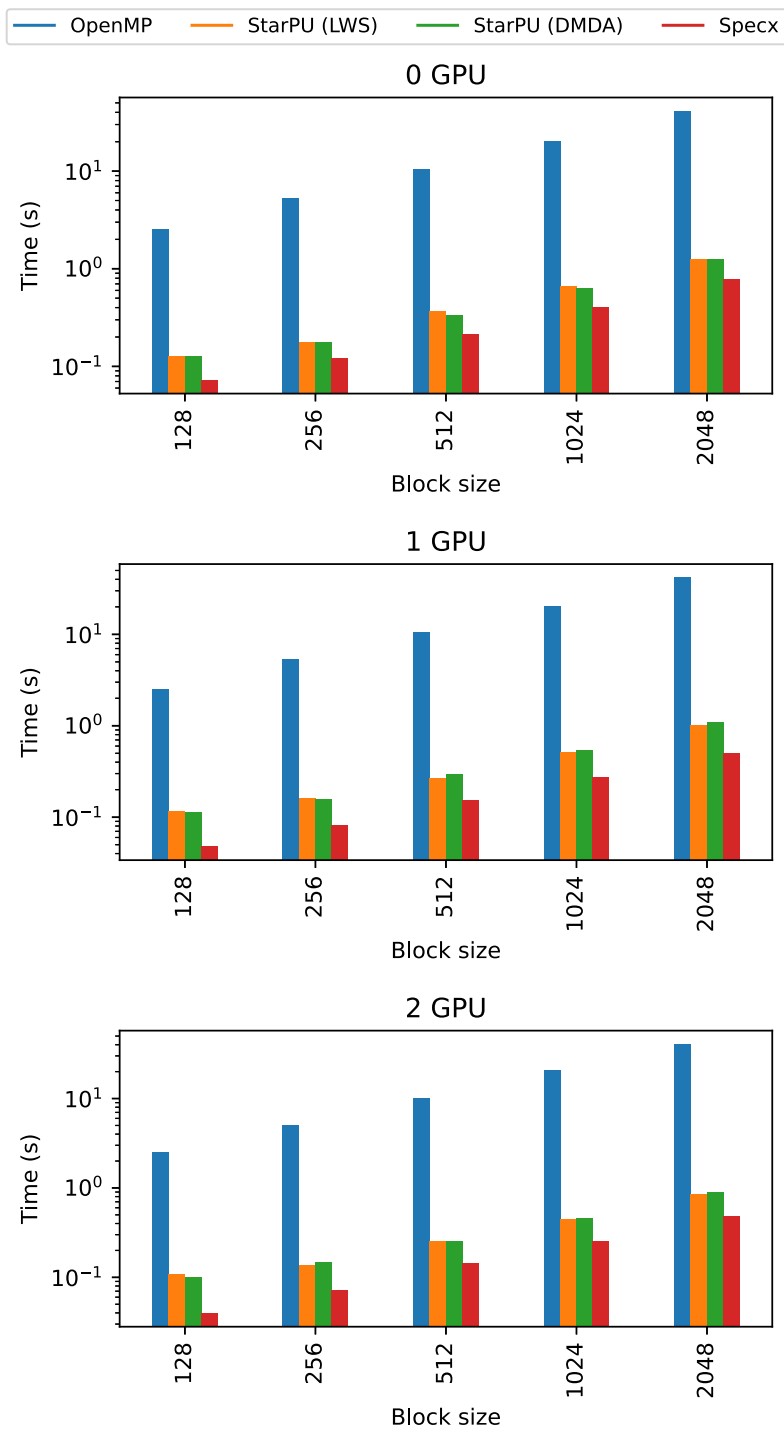

**Figure 9** Comparison of Specx against OpenMP and StarPU for the particle simulation test case on the A100 configuration.

## CONCLUSION

We presented Specx, a task-based runtime system written in C++ and for C++ applications. Specx allows parallelization over distributed computing nodes and joint harnessing of CPUs and GPUs. It is easy to use and provides advanced features such as scheduler customization and execution trace visualization. Performance study shows that Specx can be used for high-performance applications.

We plan to improve Specx by providing a scheduler made for heterogeneous computing nodes, creating new speculative execution models, adding conditional tasks and improving the compilation error handling. In addition, we would like to evaluate how C++ coroutines can be used to improve performance and/or the design of the runtime system.

### Funding

This work was supported by the Inria ADT project SPETABARU-H, and the ANR National project AUTOSPEC (ANR-21-CE25-0009). The funders had no role in study design, data collection and analysis, decision to publish, or preparation of the manuscript.

### Grant Disclosures

The following grant information was disclosed by the authors:
Inria ADT Project SPETABARU-H.
ANR National Project AUTOSPEC: ANR-21-CE25-0009.

### Competing Interests

The authors declare that they have no competing interests.

### Author Contributions

- Paul Cardosi performed the computation work, authored or reviewed drafts of the article, and approved the final draft.
- Bérenger Bramas conceived and designed the experiments, performed the experiments, analyzed the data, performed the computation work, prepared figures and/or tables, authored or reviewed drafts of the article, and approved the final draft.

### Data Availability

The Specx repository is available at GitLab: https://gitlab.inria.fr/bramas/specx (DOI 10.5281/zenodo.15288005).

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
