# Peer review of "Specx: a C++ task-based runtime system for heterogeneous distributed architectures"

_PeerJ Computer Science, doi:10.7717/peerj-cs.2966_

## Round 0.1 · original submission · Major Revisions

Dear Authors

We would like to express our gratitude for your submission. The reviewers have provided their feedback, which is now available for your perusal. We regret to inform you that your article has not been recommended for publication in its current form. However, we would like to encourage you to address the concerns and criticisms raised by the reviewers and resubmit your article once you have updated it accordingly. We would also like to recommend that you divide some paragraphs into two or more sections in order to enhance their comprehensibility and understandability.

Best wishes,

Reviewer 1 ·

Basic reporting

The paper is in general well-written but can be enhanced with the following items:
+ Related works should discuss the major differences between Specx and state-of-the-art. For instance, you mentioned "Specx uses a similar approach to SparPU", but what is the difference? Same for PaRSEC.
+ Taskflow was mentioned a few times but never discussed in related works.
+ No scheduling details. I assume this plays a critical role in heterogeneous computing environment. Are you using work-stealing or static scheduling? Or do you just build Specx on top of StarPU?

Experimental design

I don't see any comparison between Specx and other task-parallel systems. Without comparison, it's hard to justify the efficiency of Specx. At least the authors should consider comparison with the mentioned state-of-the-art, such as StarPU, HPX, Taskflow, TBB, and OpenMP.

Validity of the findings

It's an interesting work in the area of task-parallel programming system. However, the lack of comparison with state-of-the-art task-parallel systems makes it difficult to judge the quality of the work.

Reviewer 2 ·

Basic reporting

The paper reads well. I have some concerns about the comment by the authors "Regarding distributed parallelization, most runtime systems can be used with MPI [50 ]. " Yes, that is true, however, some task-bases systems use different approaches, like libfabric, LCI, and Gasnet. I would suggest they authors should mention that.

In Section "2.2 Computing on heterogeneous architectures", I recommend to mention Kokkos and SYCL to address other GPUs architectures. I would love to see a Table with all mentioned runtime systems and showcase what kind of GPUs architectures these support. I have seen similar tables in other papers.

Experimental design

The experimental design sounds solid. However, I would like to see FLOP/s reported to compare theis approach with other approaches.

Validity of the findings

Again, I would like to see FLOP/s reported to compare with other approaches.

Additional comments

I would suggest to check some of the references for HPX and other runtime systems. Some of them seem to be older and newer ones are provided on the github pages.

Reviewer 3 ·

Basic reporting

The paper "Specx: a C++ task-based runtime system for heterogeneous distributed architectures" introduces a task-based runtime system aimed at efficiently managing workloads across heterogeneous computing architectures. The system, Specx, is designed to facilitate parallelization by leveraging modern C++, allowing for dynamic workload balancing across CPUs and GPUs. The paper presents a discussion on the architecture, features, and implementation of Specx, including its ability to handle task graphs, scheduling, and speculative execution. Performance evaluations are conducted using multiple benchmarks, including GEMM, Cholesky factorization, vector-scalar products, and particle interactions, demonstrating the system's effectiveness.

Despite its strengths, the paper has some notable weaknesses. One of the main weaknesses is the lack of clarity regarding scalability testing. It appears that the experiments were conducted using only two GPUs at a time, rather than in a multi-GPU or multi-node environment. Could you clarify whether multiple GPU models (V100, A100, P100, RTX8000) were tested together in a larger heterogeneous setup, or if each test was conducted separately on different hardware? 

If the experiments were indeed limited to single-node setups with only two GPUs per test, this raises concerns about Specx’s scalability. Many HPC applications require execution across four or more GPUs, particularly in multi-node distributed computing environments, where task scheduling and memory management become significantly more complex. Without benchmarks demonstrating multi-GPU or multi-node performance, it is difficult to assess whether Specx is suitable for large-scale heterogeneous architectures. Expanding the evaluation to include multi-GPU and multi-node experiments would significantly strengthen the paper’s contributions.

Another notable limitation is the lack of real-world application benchmarks. While the authors test Specx using synthetic workloads such as GEMM, Cholesky factorization, vector-scalar products, and particle interactions, they do not demonstrate its performance in practical scientific computing applications. For example, integrating Specx into a large-scale deep learning training pipeline, computational fluid dynamics (CFD), molecular simulations, or weather forecasting would provide stronger evidence of its effectiveness in real-world scenarios. Without these practical benchmarks, it is unclear whether Specx would outperform existing runtime systems such as StarPU or OpenMP in domains where heterogeneous computing is already widely used.

Another critical issue is the complexity of Specx’s API and implementation requirements. While the paper emphasizes flexibility and modularity, the system introduces several new API components for task insertion, scheduling, and memory management, which may pose a steep learning curve for users unfamiliar with task-based parallel programming. Additionally, manual tuning appears to be required for scheduler configurations and memory transfers, making it less user-friendly compared to some existing solutions. The paper does not include a usability study, making it difficult to assess whether Specx’s performance benefits justify the additional implementation complexity.

The discussion on speculative execution is also somewhat incomplete. While the authors introduce speculative execution as a feature to increase parallelism, they do not fully explore its potential downsides. For example, speculative execution can sometimes increase computational overhead, particularly when incorrect predictions lead to unnecessary work. The paper lacks an in-depth performance breakdown of when speculative execution improves efficiency versus when it adds overhead. This omission makes it difficult to evaluate whether Specx’s speculative execution model is genuinely advantageous in practice.

Additionally, the scheduling strategy evaluation lacks depth. The paper introduces several scheduling mechanisms, such as priority-based schedulers and heterogeneous task handling, but does not extensively compare them to existing state-of-the-art scheduling techniques. For example, more advanced heuristics or machine learning-based scheduling methods could provide better task allocation in dynamic workloads. It would be valuable to see an evaluation of different scheduling algorithms under various workload conditions, especially for heterogeneous computing environments where balancing CPU-GPU workloads efficiently is crucial.

A significant issue is that the research gap and novelty of Specx are not clearly articulated. While the paper presents technical advancements, it does not clearly define what specific limitations of existing systems it overcomes. The introduction briefly discusses other task-based runtime systems such as StarPU and OpenMP, but it does not convincingly argue how Specx fills a critical gap in the field. Is Specx intended to be faster, easier to use, more scalable, or more flexible than existing solutions? The paper would benefit from a clearer justification of its contributions, explicitly demonstrating why Specx is necessary and how it advances the state of the art. Without a strong research gap, the novelty of Specx remains unclear, potentially making it difficult for readers to see its distinct advantages over existing frameworks.

Finally, the paper does not discuss potential limitations in fault tolerance or robustness. In real-world HPC applications, system failures, memory leaks, and node crashes can impact performance, and it is unclear how Specx handles such failure scenarios. A section discussing error recovery mechanisms or fault tolerance strategies would strengthen the paper’s practical contributions.

Experimental design

please see the full review in Basic Reporting.

Validity of the findings

please see the full review in Basic Reporting.

Additional comments

please see the full review in Basic Reporting.

---

## Round 0.2 · Minor Revisions

Dear Authors,

Notwithstanding the fact that two reviewers have accepted the paper, one reviewer has expressed a lack of satisfaction with the revision. It is recommended that the concerns, suggestions and criticisms made by Reviewer 3 are addressed appropriately, and that the paper is resubmitted once it has been updated accordingly.

Best wishes,

Reviewer 1 ·

Basic reporting

I am satisfied with this revision.

Experimental design

I am satisfied with this revision.

Validity of the findings

I am satisfied with this revision.

Additional comments

I am satisfied with this revision.

Reviewer 2 ·

Basic reporting

The authors addressed my previous comments.

Experimental design

The authors addressed my previous comments.

Validity of the findings

The authors addressed my previous comments.

Additional comments

The authors addressed my previous comments.

Reviewer 3 ·

Basic reporting

I am disappointed with the authors' response to my concerns regarding the paper. They did not demonstrate basic courtesy in addressing the issues I raised. For example, when I pointed out the lack of clarity regarding the scalability of the proposed approach, the authors simply stated, "On different hardware, we think this is clearly visible on the figures". I had expected a clearer explanation to be added to the manuscript.
Similarly, when I raised concerns about the usability of the tool, their response ​is, ​"We are convinced that the interface is elegant and simple​". Such a response does not engage with the substance of the comment or provide any evidence or clarification. Again, I would have expected a more constructive and informative reply, possibly supported by user testing or a discussion in the paper.

If the authors are unable to adequately respond to such concerns, they should acknowledge them as limitations of their work.

That said, I recognize that the manuscript offers valuable contributions, and I do not wish to hinder its publication. My comments were intended to help the authors better articulate the limitations and improve the overall clarity of the manuscript. If the editor deems the authors​' response satisfactory, I have no objection to the manuscript being accepted.

Experimental design

please see my response to 1. Basic Reporting

Validity of the findings

please see my response to 1. Basic Reporting

Additional comments

please see my response to 1. Basic Reporting

---

## Round 0.3 · accepted · Accept

Dear Authors,

Thank you for addressing the reviewer's concerns. Your manuscript now seems sufficiently improved and ready for publication.

Best wishes,

Reviewer 3 ·

Basic reporting

I am disappointed by the current situation. I have volunteered my time and expertise to review this paper as a service to the academic community, without any expectation of personal benefit. My intention throughout the review process has been constructive. While I raised several important concerns, my comments were clearly aimed at improving the paper through clarification and refinement, not rejecting it. If I had intended to recommend rejection, I would have done so outright during the first round. Instead, I recommended major revisions initially, and, despite receiving responses from the authors that were dismissive and discourteous, I generously opted for a minor revision in the subsequent round.

Now, the authors have responded by making unfounded allegations, suggesting that my review was generated by an AI language model, without providing any evidence. I must ask: are my comments not valid, reasonable, and academically sound? Rather than deflecting the issues raised, would it not be more productive for the authors to simply clarify the experimental setup, address the questions, and acknowledge the limitations accordingly?

For example, if the scalability evaluation does not include multi-GPU or multi-node environments, why not state this in the limitations? If the usability of the system has not been evaluated, why not acknowledge it and indicate it as an area for future work? These clarifications would not only improve the transparency of the paper but also strengthen its contribution by highlighting opportunities for further research.

Despite the persistent disrespectful tone of the authors and their failure to provide a professional and evidence-based response to my review, I am again overlooking and recommending acceptance of this paper at this stage. I do ask the authors to respond to my concerns by providing clarification, such as by adding these to the limitations and future work. I leave this to the authors and the editor to decide; in the end, this is supposed to improve their paper.​

​I advise the authors to avoid such behavior in the future. How would they feel if, after sincerely writing a review to help improve a paper, they received a response like the one they had given?

Experimental design

This is the second revision of the manuscript. I refer the authors to my previous comments, as well as the additional points outlined in Section 1 of this review.

Validity of the findings

This is the second revision of the manuscript. I refer the authors to my previous comments, as well as the additional points outlined in Section 1 of this review.

Additional comments

This is the second revision of the manuscript. I refer the authors to my previous comments, as well as the additional points outlined in Section 1 of this review.